. Pathogens

# Determinants of natural killer cell-mediated antibody dependent cellular cytotoxicity in SARS-CoV-2 antibodies

Delphine M. Depierreux[1], Felicitas Ruiz[1], Michelle Lilly[1], Jamie Guenthoer[1], Vrasha Chohan[1], Julie Overbaugh [1,2]*

1 Division of Human Biology, Fred Hutchinson Cancer Center, Seattle, Washington, United States of America, 2 Public Health Sciences Division, Fred Hutchinson Cancer Center, Seattle, Washington, United States of America

* joverbau@fredhutch.org

## Abstract

A growing body of evidence underscores the role of antibody-dependent cellular cytotoxicity (ADCC) in antiviral immunity. Yet, the mechanisms underlying the ability of certain antibodies (Abs) to mediate potent ADCC activity remain poorly characterized. In particular the contribution of features within the antigen-binding Fab region remains largely unexplored. In this study, we leveraged a collection of 142 SARS-CoV-2 monoclonal Abs to systematically dissect the determinants of ADCC activity. We analyzed their epitope domain target, binding characteristics, neutralization potency, somatic hypermutation (SHM) and CDR3 length to determine the contribution of these features to ADCC activity. We found that ADCC activity is primarily shaped by epitope target - particularly targeting of the S2 domain of the Spike glycoprotein. ADCC potency was not associated with the degree of SHM or neutralization. Notably, ADCC activity was not correlated with binding activity and moderate binding to antigen was sufficient for ADCC activity. By integrating these analyses, we provide a comprehensive framework for understanding the molecular and functional determinants of ADCC. Together, these findings offer novel insights into the mechanisms that underpin ADCC functions, with implications for vaccine design and therapeutic Abs development.

## Author summary

While antibodies are best known for stopping viruses from entering cells, they also activate other arms of the immune system. One important function is their ability to signal immune cells to destroy virus-infected targets, a process called antibody-dependent cellular cytotoxicity (ADCC). In this study, we examined 142 antibodies directed against the Spike protein of the virus that causes COVID-19. We compared their binding patterns, neutralizing ability, level of maturation,

**Data availability statement:** All data are in the manuscript and/or supporting information files.

**Funding:** This work has been supported by the National Institutes of Health (www.nih.gov), through grants AI38709to J.O. and a Washington Research Foundation Postdoctoral Fellowship to D.M.D (www.wrfseattle.org). The funders had no role in study design, data collection and analysis, decision to publish, or preparation of the manuscript.

**Competing interests:** The authors have declared that no competing interests exist.

and other features to determine what drives their capacity to trigger ADCC. We found that the region of the Spike protein targeted by the antibody was the main factor shaping this activity, with antibodies that recognized the S2 region being particularly effective. In contrast, the amount of antibody maturation, the strength of binding, or the ability to neutralize the virus did not predict immune cell activation. Our findings show that antibodies can contribute to antiviral defense through distinct mechanisms and provide new insights for guiding vaccine and antibody therapy design.

## Introduction

Antibody-dependent cellular cytotoxicity (ADCC) is a key immune defense mechanism that eliminates virus-infected cells. This process is initiated when antibodies (Abs) bind to viral antigens on the surface of infected cells via their fragment antigen-binding (Fab) regions, while the fragment crystallizable (Fc) regions engage with Fc receptors (FcRs) expressed on effector cells [1]. Natural killer (NK) cells are the primary mediators of ADCC, acting through the CD16 (FcγRIII) receptor, which binds the Fc portion of IgG Abs [2–4].This Fc-FcR interaction triggers the formation of an immune synapse and the release of cytotoxic granules, ultimately leading to target cell apoptosis [5]. While the Fc-FcR interaction is well-characterized, less is understood about how Fab domain interaction with the antigen influences ADCC.

ADCC has been implicated in protection against a range of viral infections, including HIV, influenza, and Ebola, both following natural infection and vaccination [6–16]. In the context of SARS-CoV-2 infection, ADCC has emerged as a correlate of protection in both animal models and human studies [17,18]. ADCC-mediating Abs have been detected after natural infection and vaccination [19–22]. Notably, individuals with hybrid immunity to SARS-CoV-2 (vaccination and infection) have been shown to mount broader and more potent Abs responses, including enhanced Fc-mediated effector functions, compared to those with infection or vaccination alone [21,23]. ADCC has been shown to contribute to viral clearance and delayed spread in animal models [17,24]. In patients with severe COVID-19, higher ADCC activity correlates with better outcomes [18,25]. Furthermore, therapeutic monoclonal Abs such as Sotrovimab (S309) have demonstrated *in vivo* efficacy through both neutralization and Fc-dependent functions [26–28]. Although it is becoming clear that ADCC-mediating Abs are a correlate of protection and have potential as therapeutics, the study of the determinants of ADCC Abs activity has been much more limited than studies of neutralizing Abs.

Abs to the Spike glycoprotein, which is the SARS-CoV-2 antigen in the COVID-19 vaccine, can have either neutralizing or non-neutralizing activity or both, with the majority (95%) being non-neutralizing [29]. Spike is a trimer of heterodimers comprised of two subunits, S1 and S2. The S1 subunit contains an N-terminal domain (NTD), a receptor-binding domain (RBD), and the C-terminal subdomain (called CTD or SD). While neutralizing Abs typically target the receptor-binding domain (RBD)

or the N-terminal domain (NTD) of Spike and act by blocking the interaction with the angiotensin-converting enzyme 2 (ACE2) receptor, their durability is often limited by Spike mutations found in emerging SARS-CoV-2 variants [30–34]. In contrast, ADCC-mediating Abs, which have been detected up to 400 days post-infection, tend to retain activity across variants, underscoring the importance of Fc-effector mechanisms in durable protection [18,35–37]. Yet, the molecular determinants of ADCC activity remain incompletely understood. Previous studies have identified Fc domain features-such as glycosylation patterns and engineered mutations (e.g., GASDALIE) - that enhance FcγRIIIa binding and ADCC activity [14,17,24,38–41]. Other studies have highlighted that the epitope bound by Abs can influence the potency of FcR activation, as shown in the context of influenza and HIV [42].However, the influence of Fab-region characteristics on ADCC function has received relatively little attention and the factors that make Abs capable of mediating ADCC are not well defined.

In this study, we aimed to define the factors that drive strong ADCC activity in Spike-specific SARS-CoV-2 Abs. Using a collection of 142 Spike-specific Abs isolated from a donor (C68) with hybrid immunity [43–46] we systematically assessed ADCC function, epitope domain target, binding properties, neutralization and SHM and CDR3 length. Through this integrated analysis, we provide new insights into the Fab features that contribute to effective ADCC.

## Results

### ADCC Abs tend to target S2 whereas neutralizing Abs more often target RBD

To assess whether certain domains of the SARS-CoV-2 Spike protein are preferentially targeted by ADCC-mediating Abs, we mapped the epitope domain of 142 Abs isolated from a single individual at two time points after they experienced post-vaccination infections (PVI) [43–46]. As shown in Fig 1A-1B, most Abs bound to the RBD (n = 69) or S2 (n = 45), with fewer targeting the NTD (n = 22) or CTD (n = 6).

Each Abs ability to trigger NK cell-mediated ADCC was evaluated against Spike-expressing target cells. The majority of S2 (27/45, 60%) and CTD-specific Abs (5/6, 83%) demonstrated detectable ADCC against the ancestral Wuhan WH-1 (D614G variant) Spike protein (Fig 1C), a trend also observed for Omicron variants BA.2 and BA.5 with the highest percentage of ADCC-mediating S2 Abs against BA.5 (38/45, 84%; S1C Fig). Relatively fewer RBD or NTD-specific Abs showed ADCC activity (22/69, 32% and 4/22, 18%, respectively) (Fig 1C). Likewise, S2-specific Abs mediated significantly stronger ADCC responses compared to those targeting the RBD or NTD (Fig 1D), a pattern consistent across BA.2 and BA.5 Spike variants (S1B-S1D Fig), indicating this effect is conserved across variants of concern (VOCs). Fewer Abs showed ADCC activity against BA.2 Spike protein compared to the other two antigens. Across S2-specific Abs, each epitope group, as defined by competition ELISA in a previous publication [45], contained at least one antibody capable of mediating ADCC. Notably, ADCC activity did not differ significantly across epitope groups for D614G, BA.5, or BA.2 Spike (Kruskal-Wallis p = 0.7109, 0.2041, and 0.6721, respectively) (S2 Fig).

In contrast, neutralizing Abs primarily targeted the RBD (61/69) and CTD (5/6) domains (Fig 1E), with no S2- and only one NTD-directed Abs showing neutralizing activity (Fig 1E). RBD and CTD Abs were most potent neutralizers (Fig 1F). Together, these findings indicate that ADCC-mediating Abs recognize a broader range of epitopes than neutralizing Abs and that those targeting the S2 domains are more likely to induce potent ADCC. In the case of the CTD Abs, few were analyzed and three were highly related members of the same clonal family, limiting our ability to draw conclusions on the CTD epitope.

### Neutralization and ADCC functions are independent

Some SARS-CoV-2-specific Abs can mediate both neutralization and ADCC [26,27,47]. However, it remains unclear whether these two functions interact in a way that enhances or impairs one another. To address this, we compared the ADCC activity of neutralizing and non-neutralizing Abs against WH-1 SARS-CoV-2 (Fig 2A). We observed no significant

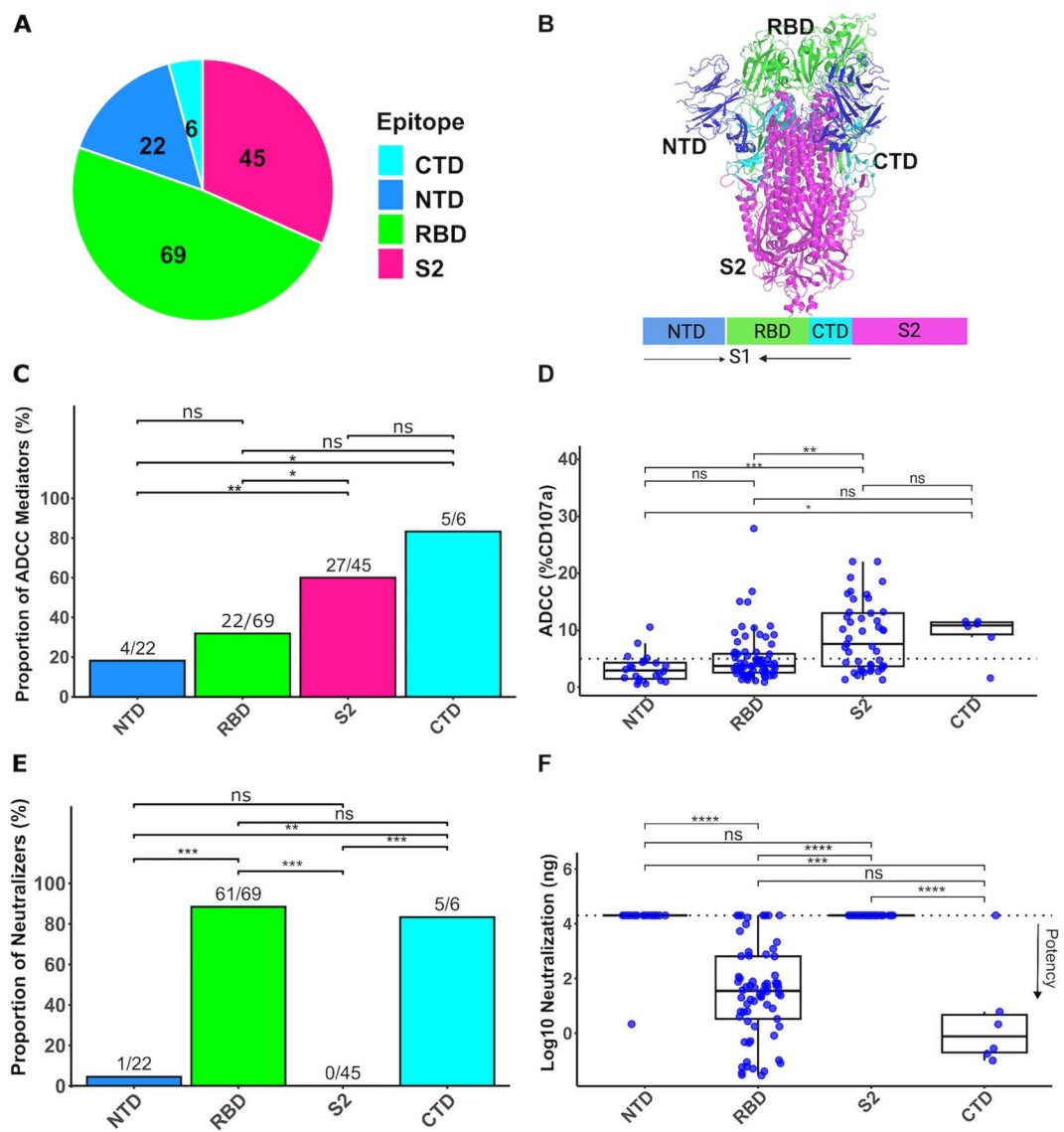

**Fig 1. Epitope specificity and functional activity of C68 Abs.** (A) Distribution of C68 Abs across SARS-CoV-2 Spike protein domains. Pie chart segments indicate total Abs targeting each domain: NTD (22), RBD (69), S2 (45), and CTD (6). (B) Schematic of full-length Spike and cryo-EM structure of the WH-1 SARS-CoV-2 Spike trimer (PDB: 6VSB), with domains color-coded to match the pie chart and bar graph designations NTD (blue), RBD (green), CTD (cyan), S2 (magenta); adapted from [59]. (C) Proportion of ADCC-capable Abs across domains of the WH-1 SARS-CoV-2 spike protein. The numerical fractions above each bar indicate the number of ADCC capable Abs relative to the total number of Abs against the indicated domain. Abs were defined as ADCC-capable if the average %CD107a+NK cells from two independent experiments exceeded 5%, as described in methods. Pairwise comparisons were performed using Fisher exact test with Holm correction for multiple testing. Of note, CTD Abs are clonally restricted, limiting our ability to draw conclusions. (D) ADCC potency of Abs by Spike domain. Each point represents a single Ab. Boxplots show median, IQR, and 1.5×IQR whiskers. Dotted line indicates limit of detection. Wilcoxon tests with Holm correction; significance indicated as $*p < 0.05$, $**p < 0.01$, $***p < 0.001$, $****p < 0.0001$; ns = not significant. (E) Proportion of Abs that neutralize pseudovirus bearing the WH-1 SARS-CoV-2 Spike protein shown by epitope domain target. The numerical fractions above each bar indicate the number of neutralizing Abs relative to the total number of Abs against the indicated domain. Neutralization status was considered positive when $IC_{50} < 20,000$ ng/mL as described in methods. Pairwise comparisons were performed using Fisher exact test with Holm correction for multiple testing. F) Neutralization potency ($\log_{10}[IC_{50}$, ng/mL]) by Spike domain. Dotted line indicates limit of detection. Wilcoxon tests with Holm correction as in (D).

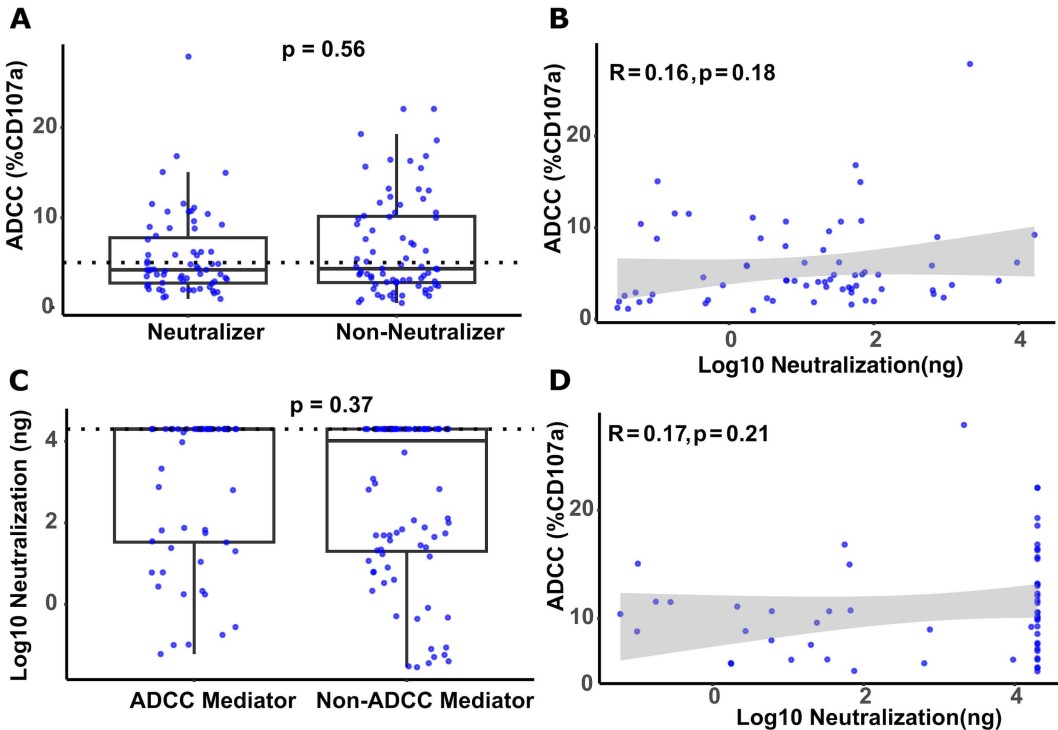

**Fig 2. Relationship between ADCC and neutralization activity against WH-1 SARS-CoV2.** (A) ADCC potency for neutralizing vs. non-neutralizing Abs. Two-sided Wilcoxon rank-sum test p value is shown. Dotted line indicates limit of detection. (B) Correlation between neutralization potency and ADCC activity across all Abs. Pearson R, p-value, and 95% CI (gray shading) from linear regression shown. (C) Neutralization potency of ADCC-mediating vs. non-mediating Abs. Two-sided Wilcoxon rank-sum test p value is shown. Dotted line indicates limit of detection. (D) Correlation between neutralization and ADCC activity among ADCC-mediating Abs. Pearson correlation coefficient (R), p-value, and 95% CI (gray) are shown. In all boxplots, each point represents a unique mAb; boxes indicate the IQR, the line marks the median, and whiskers extend to 1.5 × IQR.

difference between the two groups (p = 0.56), suggesting that neutralization status does not affect ADCC potency. Consistent with this, among neutralizing Abs, we found no correlation between neutralization and ADCC activity (Fig 2B; R = 0.16, p = 0.18). A complementary analysis comparing the neutralization potency of Abs that mediate ADCC to those that did not, also showed no significant difference (Fig 2C; p = 0.37), indicating that ADCC activity does not affect neutralization potency. Additionally, within the group of ADCC-mediating Abs, there was no correlation between ADCC and neutralization potency (Fig 2D; R = 0.17, p = 0.21). Together, these data suggest that neutralization and ADCC functions of these Spike-specific Abs are independent.

## Binding to cell-surface expressed spike at a fixed concentration does not predict ADCC magnitude

To explore the contribution of antigen binding to ADCC activity, we assessed all 142 Abs for their ability to bind the WH-1 Spike protein expressed on target cells by flow cytometry (gating strategy shown in S3A Fig). We found no significant correlation between binding activity (median fluorescence intensity (MFI)) and ADCC activity for the WH-1 (Fig 3, R = 0.04, p = 0.64) or Omicron BA.2, and BA.5 Spike proteins although there was a non-significant trend for correlation with the BA.5 Spike antigen (S3B Fig, R = -0.089, p = 0.47; S3C Fig, R = 0.21, p = 0.066). These findings overall indicate that while a minimum level of antigen binding is necessary for ADCC, the magnitude of the ADCC response largely varies independently of binding to the antigen.

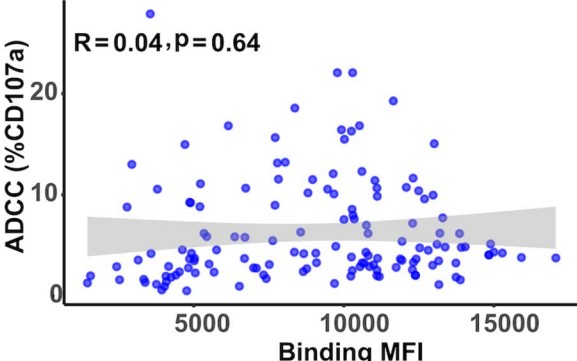

**Fig 3. Relationship between antigen binding and ADCC activity against WH-1 SARS-CoV-2 Spike.** Correlation between binding to WH-1 SARS-CoV-2 Spike (MFI) and ADCC potency (% CD107a+ NK cells) for binder Abs (MFI > 1000). Each data point represents individual Abs. Pearson correlation coefficient (R), p value and 95% CI (gray) are shown.

## SHM has limited impact on ADCC activity compared to neutralization

To explore whether Abs maturation influences ADCC, we quantified SHM in the heavy chain (VH) and light chain (VL) of all 142 Abs from the reconstructed repertoire. Our results showed that the percentage of SHM in the VH was higher in ADCC mediators (Fig 4A, p = 0.033) while SHM in the VL was comparable between Abs that mediate ADCC and those that do not (Fig 4B p = 0.32), with both groups exhibiting a similar distribution. However, within the subset of ADCC-mediating Abs, there was no correlation between %SHM and ADCC activity (Fig 4C and 4D) for both the VH and VL (R = -0.059, p = 0.66; R = -0.071, p = 0.59). In contrast, among neutralizing Abs, we observed a significant positive correlation between %SHM and neutralization potency (S4A and S4B Fig) for both the VH (R = -0.54, p = 1.9e-6) and VL (R = -0.39, p = 0.0013). Together, these findings indicate that, unlike neutralization, SHM, which drives Abs maturation, is not a key determinant of ADCC.

## Additional antigen exposure is not associated with ADCC activity

The C68 Abs were isolated from two longitudinal time points, spanning additional antigen exposures through vaccination and two post-vaccination infections. Segregating the Abs per time point allowed us to assess the impact of repeated antigen exposure on ADCC and neutralization. Fig 5A shows the distribution of Abs targeting each epitope domain (NTD, RBD, S2, CTD) at post-vaccination infection 1 (PVI-1; n = 99) and post-vaccination infection 2 (PVI-2; n = 43). Abs isolated at the later time point exhibited significantly enhanced neutralization of WH-1 SARS-CoV-2 (Fig 5B, p = 0.00015), while ADCC activity remained unchanged for both WH-1 SARS-CoV-2 (Fig 5C, p = 0.23) and Omicron BA.5 (S5D Fig, p = 0.32). A lower ADCC activity was observed against BA.2 at PVI-2 compared to PVI-1 (S5C Fig, p = 5.3 e-05). There was a significant increase in SHM, both in the VH (S5B Fig, p = 2.6 e-14) and VL (S5A Fig, p = 2.3 e-08). Taken together, these results suggest that repeated antigen exposure is associated with enhanced neutralizing activity and higher SHM levels, while ADCC functionality remains largely unchanged.

## CDR3 length does not influence ADCC activity

While the CDR3 region of Abs is critical for antigen specificity, its role in ADCC remains unclear. Using our sequencing data, we analyzed the correlation between ADCC or neutralization and CDR3 length for both the VH and VL. Among ADCC-mediating Abs, no significant correlation was observed between CDR3 length and ADCC activity for either Abs chains (Fig 6A, R = 0.057, p = 0.67; Fig 6B, R = 0.13, p = 0.32). A similar analysis among neutralizing Abs also showed no

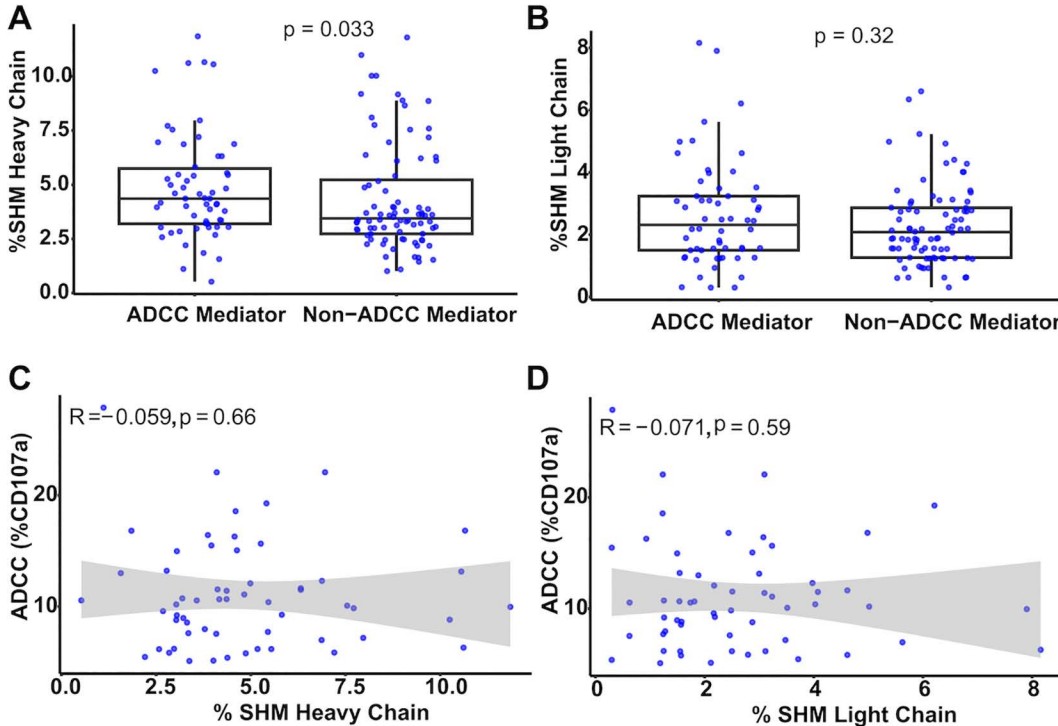

**Fig 4. Relationship between ADCC activity and antibody maturation.** (A–B) Comparison of %SHM in the VH (A) and VL (B) chains between ADCC Mediator and Non-ADCC Mediator Abs as defined in methods. Each point represents one mAb. P-values from Wilcoxon rank-sum tests are shown. (C–D) Correlation of ADCC activity with %SHM in VH (C) and VL (D) among ADCC-mediating Abs. Pearson correlation coefficients (R), p-values and 95% CI (gray shading) are shown.

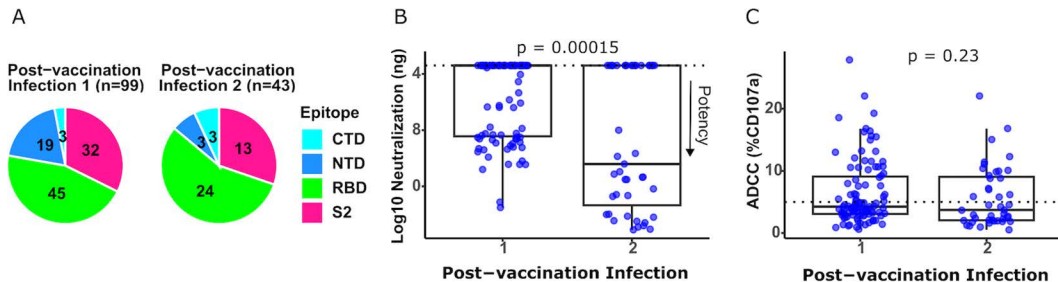

**Fig 5. Longitudinal comparison of mAb function against the WH-1 SARS-CoV2 Spike protein.** (A) Pie charts showing the distribution of C68 mAb across SARS CoV-2 Spike domain (NTD, RBD, S2, CTD) at post-vaccination infection 1 (PVI-1) and PVI-2. (B–C) Boxplots comparing (B) neutralization potency (log₁₀ IC50 in ng/mL), and (C) ADCC activity between Abs isolated at PVI-1 and PVI-2. Each dot represents one mAb. Dotted line indicates limit of detection. Statistical comparisons were performed using the Wilcoxon rank-sum test; p-values are shown above each plot.

correlation (S6A Fig, R = 0.13, p = 0.29, S6B Fig, R = 0.031, p = 0.8). These findings suggest that CDR3 length is not a major determinant of ADCC or neutralization functionality against SARS-CoV-2.

## Discussion

ADCC has emerged as a critical immune effector mechanism against viral infections, including SARS-CoV-2. Despite growing interest, the molecular determinants of ADCC potency remain poorly understood - particularly compared to the

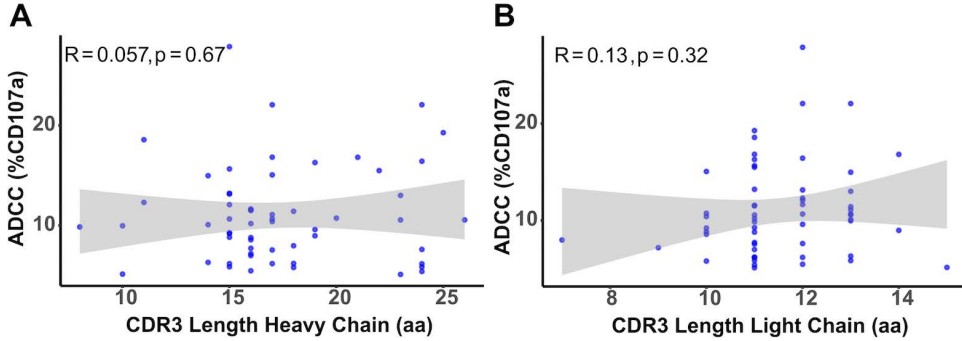

**Fig 6. Correlation between mAb CDR3 length and ADCC activity.** Scatter plots showing the correlation between CDR3 length (VH (A), VL (B) and ADCC activity against WH-1 SARS-CoV-2. Analysis includes ADCC capable Abs. Pearson correlation coefficient (R), p-value, and 95% CI (gray shading) are shown.

extensive knowledge surrounding neutralizing Abs. While Fc receptor engagement is required for ADCC, the contribution of Fab-region features has received limited attention. In this study, we leveraged a diverse panel of 142 SARS-CoV-2 Abs to identify molecular features associated with ADCC activity. Our findings reveal that ADCC activity is primarily determined by the targeted epitope domain, with increased potency for Abs targeting the S2 domains while neutralization relies more heavily on affinity maturation and binding to variable regions like the RBD. Importantly, these functions appear to evolve independently.

Our data showing that Abs targeting the S2 domain are significantly more likely to mediate potent ADCC compared to those binding the RBD or NTD, are consistent with prior reports demonstrating that depletion of serum Abs against S2 - but not RBD or NTD - most markedly reduces ADCC activity [21,48,49]. Our data suggest that ADCC activity is distributed broadly across the S2 domain and is not restricted to a single S2 epitope group. Similarly, the observation that ADCC Abs recognized a broader set of Spike epitopes than neutralizing Abs, which were largely restricted to the RBD, is consistent with prior reports [32,48,50]. The spatial location of the S2 domain may explain the preference for this target: epitopes closer to the cell membrane, such as those in S2 domain, may be more effective at clustering Fc receptors and stabilizing immune synapses to enhance ADCC [51]. Notably, regions such as the S2 domain are more conserved compared to the RBD and NTD. Multiple studies have demonstrated that Fcγ receptor engagement by Abs remains largely preserved across VOCs, despite substantial declines in neutralization titers [29,37,52]. Consistent with these observations, we found that S2-specific Abs robustly mediated ADCC across diverse variants, including the Omicron sublineages BA.2 and BA.5, although we noted that a higher percentage of S2 Abs mediated ADCC against BA.5. Notably, other work has similarly reported that S2-specific mAbs mediate potent ADCC that resist BA.1 escape and *in vivo* studies have shown that S2 mAbs confer protection through Fc-mediated functions such as ADCC [17,26,49,53].

Another insight from our data is that binding did not correlate with ADCC activity. While binding was necessary for ADCC activity, Abs with comparable binding to Spike-expressing cells showed markedly different ADCC responses. This mirrors observations in HIV where clonally related Abs with comparable binding affinity can exhibit markedly different ADCC potencies [54]. This dissociation implies that qualitative features of epitope recognition-such as location, accessibility, or angle of approach - may be more critical for triggering ADCC than overall antigen binding activity. These findings also have implications for identifying optimal therapeutic Abs as selecting the strongest antigen binders may not necessarily identify the most effective ADCC mediators.

By examining both neutralization and ADCC for this large collection of Abs, we found that these activities are functionally independent. Neutralizers and non-neutralizers were equally likely to mediate ADCC, and neutralization potency did not predict ADCC. While SHM is a well-established driver of neutralization breadth and potency - particularly in HIV-1

[55,56] - we observed only a modest increase in SHM in the VH in ADCC-mediating Abs compared to non-mediators. Within ADCC-mediating Abs, SHM levels did not correlate with ADCC potency, suggesting that beyond a minimal threshold, further somatic diversification does not enhance ADCC. SHM in the VL was comparable between groups, reinforcing the idea that extensive affinity maturation is not required for ADCC.

Moreover, while antibodies isolated after repeated antigen exposure via vaccination and post-vaccination infection in this individual showed increased SHM and enhanced neutralizing activity, they did not show improved ADCC function. Notably, mAbs isolated after the second post-vaccination infection have reduced ADCC against BA.2 compared with the first infection, but no difference in ADCC against BA.5. Because the second post-vaccination infecting virus was not sequenced, we cannot attribute this pattern characteristics of the infecting variant; however, one plausible explanation is that post-vaccination infection 2 preferentially boosted BA.5-biased specificities (or shifted epitope targeting) that preserved BA.5 recognition while reducing cross-reactive engagement of BA.2. Although CDR3 length is critical for HIV-1 neutralization, we found no correlation between CDR3 length and ADCC activity in SARS-CoV-2 Abs, suggesting it is not a key determinant of ADCC. Our finding suggest that ADCC-competent Abs may emerge earlier in the immune response, without requiring extensive affinity maturation. This could be particularly advantageous during primary infection or in immunocompromised individuals, who may have limited capacity to generate highly mutated, high-affinity neutralizing Abs.

Taken together, our findings support a model in which ADCC is predominantly determined by epitope targeting, particularly to conserved, membrane-proximal regions of Spike such as S2, while neutralization relies more heavily on affinity maturation and binding to variable regions like the RBD. Our work provides a foundation for efforts to enhance Abs functions in the broader landscape of antiviral immunity.

## Limitations

The Abs evaluated in this study are derived from a single donor, which limits generalizability. The relative abundance and functional dominance of specific Spike domain Abs may reflect donor-specific immunodominance shaped by exposure sequence, repertoire constraints, germline gene usage, and potentially host genetics. Confirming whether these patterns hold more broadly will require systematic multi-donor studies. This study presents results based on the broad domains of Spike and higher-resolution epitope mapping will be required to further refine epitope-level determinants of activity. Binding was assessed at a single concentration without kinetic measurements, precluding strong inferences about affinity/avidity or kinetic contributions. Our functional readouts quantify NK-cell activation rather than direct cytotoxicity and should be interpreted as effects on activation potential and complemented by direct killing/target-cell loss assays in future work. Fc glycosylation is an additional variable that can modulate FcγR engagement and NK-cell effector function. Although in this study, care was taken to avoid batch effects that could impact findings, future studies should consider incorporating dedicated Fc glycan-profiling to explore the impact of Fc glycosylation.

## Conclusion

Together, our findings highlight the importance of epitope targeting in shaping ADCC function and demonstrate that Fab-region features can strongly influence Fc-mediated effector responses. Our study reveals that ADCC activity against SARS-CoV-2 Spike is primarily driven by epitope specificity, particularly targeting the membrane-proximal S2 domain, while not being associated with antigen binding activity. Unlike neutralizing Abs, which are often highly mutated and target domains such as the RBD, ADCC-mediating Abs target a broader array of Spike epitopes, including regions less prone to mutation. This broader specificity and independence from affinity maturation suggest that ADCC-capable Abs can arise early in the immune response and remain effective across evolving variants. This study is the first to shed light on Fab characteristics that influence ADCC activity, offering important insights for the development of vaccines and therapeutics that leverage both neutralizing and Fc-mediated effector functions.

## Methods

### Source of Abs

SARS-CoV-2 memory B cells from subject C68 were isolated with previously described standard methods and Abs were reconstructed using standard methods in FreeStyle 293-F Cells (Invitrogen) as described [43–46]. One hundred and forty-two Abs that bound the Spike protein were isolated from C68. The majority (n = 99) were isolated one month after a first post-vaccination infection with the Delta variant (post-vaccination infection 1, PVI-1), which was two months after immunization with two doses of the Pfizer-BioNTech mRNA vaccine, and includes all Abs isolate and expressed a functional Abs that bound Spike protein as described [43–45].Another 43 were isolated from the same individual about one year later, one month after a second post-vaccination infection (PVI-2) at a time when Omicron variant BA.2 or BA.5 were the dominant circulating strains [46], which was 8 months after a third vaccination. For the second time point (PVI-2), the Abs selected for study were based on reconstructing an Ab from a sequence representative of each VH gene family identified, generally focusing on those with the highest level of SHM within the family. All VH sequences were cloned into a common IgG1 expression vector containing the same Fc region, enabling the study to focus exclusively on Fab-mediated effects. The post-vaccination infection viruses from this participant were not sequenced, preventing definitive assignment of the infecting variant. Surveillance data indicate that BA.2 and BA.5 were circulating during this period, with BA.5 predominating. Abs amino acid sequences are provided in S2 Table.

### Abs sequence annotations and characterization

Partis software package was used to determine germline inference, clonality, levels of somatic hypermutation, and Abs CDR3 length (https://github.com/psathyrella/partis/ and https://arxiv.org/abs/2203.11367).

### Abs target domain mapping

Binding of mAbs to recombinant SARS-CoV-2 Spike subunits was assessed by direct ELISA, as previously described [43]. Epitope domains were assigned for 142 mAbs based on binding to recombinant Spike subdomains: S1, NTD, RBD, and S2. CTD specificity was inferred for antibodies binding S1 but not NTD or RBD. The proteins used included: SARS-CoV-2 S1 WH-1 strain (Sino Biological, cat. 40591-V08H), SARS-CoV-2 S2 (mammalian cell-derived, cat. 40590-V08H1), SARS-CoV-2 NTD subunit (Sino Biologicals, 40591-V49H) and SARS-CoV-2 RBD (Sino Biological, cat. 40592-V08H). S2 epitope groups were assigned using a competition ELISA approach as described previously [45]. Data were analyzed using GraphPad Prism v10.

### NK cell-mediated ADCC

We adapted a validated SARS-CoV-2 Spike-specific ADCC assay [44,45,55] to quantify NK cell activation via CD107a surface expression. Cryopreserved PBMCs (Bloodworks) were rested overnight in RPMI (Gibco) supplemented with 10% FBS (Gibco), 4.0 mM Glutamax (Gibco), and 1% antibiotic-antimycotic (Life Technologies). We used the SARS-CoV-2 Spike protein representing the WH-1 strain carrying the D614G mutation, which is referred to as WH-1 Spike throughout the manuscript. PBMCs were then mixed with CEM.NKr target cells stably expressing full-length, GFP-tagged SARS-CoV-2 Spike (WH-1, BA.2, BA.5 variants; kindly provided by Andrés Finzi) at a 10:1 effector-to-target ratio (1M: 0.1M). In a 96-well V-bottom plate, we added C68 Abs (5000 ng/mL-the highest concentration before prozone, as determined previously [45,55]) and anti-CD107a-APC (clone H4A3, BioLegend, 1:50), adjusting the volume to 100 µL. Plates were incubated at 37°C, 5% $CO_2$ for 4 hours, with a protein transport inhibitor cocktail (1:500, eBioscience) added 30 minutes in to prevent re-internalization of CD107a. Following incubation, cells were washed and stained on ice for 30 minutes with viability dye (eFluor 780, Thermo Fisher Scientific), anti-CD3-BV711 (clone UCHT1, BioLegend), anti-CD56-BV605

(clone 5.1H11, BioLegend). After washing and fixation, events were acquired on a Fortessa X50 LSR instrument (BD Biosciences). ADCC was quantified as the percentage of NK cells (live, singlets, Ef450-, CD3-, CD56 + cells) positive for CD107a (gating strategy in S7A Fig). The background ADCC signal - measured in control samples containing only effector and target cells without candidate Abs, run in triplicate - was subtracted from the values obtained for each Abs. We performed two independent repeats for each assay which showed robust reproducibility as indicated by high correlation (S7C Fig) and were averaged. We included positive controls Spike-specific Abs previously shown to mediate ADCC, CV3-25-WT and CV3-25-GASDALIE [24] and the previously FDA-licensed therapeutic S309 [27] which induced robust NK cell activation (18%, 62% and 19% CD107a+ expression, respectively; S7B Fig). The negative control Abs, A32 (HIV-specific mAb) [56] and S2X259 (SARS-CoV-2 mAb known to lack ADCC activity) [44,57],were used to establish the assay background threshold at 5% (S7B Fig). The data from two independent experiments were averaged, and Abs were considered ADCC-capable if the mean CD107a+ response exceeded 5%. Data were analyzed using FlowJo v10.7.1.

## Binding to SARS-CoV-2 Spike protein

A flow cytometry-based approach was used to examine the ability of C68 Abs to bind to SARS-CoV-2 Spike protein as described previously [45]. The conditions of this assay were used to be consistent with the ADCC assay. Briefly, 0.1M Spike-expressing cells (CEM.NKr CCR5 + cells stably expressing GFP-tagged Spike protein D614G, BA.2 and BA.5) were incubated with C68 Abs at 5000ng/mL in a final volume of 100uL in 96 V-well plate. Cells were incubated for 30 min at 37$^\circ$C, 5%CO$_2$, then washed with PBS twice, and stained with Viability dye (ef780, Bioscience) and anti-human IgG, Fcγ fragment-specific PE (Jackson Immuno, cat. 109-115-098) for 30 min. After washing and fixation, events were acquired on a Fortessa LSRII instrument (BD Biosciences) using BD FACSDiva software. Data were analyzed using FlowJo v10.7.1 (TreeStar, gating strategy shown S3 Fig). Each experiment included a positive control (CV3–25 GASDALIE) and a negative control (HIV-specific A32). Each experiment was repeated twice independently, and antibodies were considered binders if the average MFI exceeded 1000.

## Neutralization

In vitro Spike-pseudotyped neutralization assays were performed as previously described [43–45]. The virus tested was the early circulating WH-1 strain, which is identical to the Spike used for ADCC assays with the exception of the D614G mutation, which has minimal impact on epitope recognition [58]. Neutralization assays were done in technical duplicate and replicate experiments were averaged. The fraction of infectivity was calculated using GraphPad Prism by fitting a four-parameter (agonist vs response) nonlinear regression curve with the bottom fixed at 0, the top constrained to 1 and HillSlope > 1. Neutralizers were defined based on $IC_{50} < 20{,}000$ ng/mL.

## Statistics

All statistics were done using GraphPad Prism v10 and R v 2024.04.1 + 748 softwares. For pairwise comparison, data distribution normality was tested with Shapiro test. Correlation analyses used Pearson correlation coefficient because our primary interest was in quantifying linear associations, and scatterplots were inspected to confirm absence of influential outliers

## Supporting information

**S1 Fig. ADCC activity of C68 individual Abs across domains of the SARS-CoV-2 Spike protein Omicron variant.** (A, C) Proportion of ADCC-capable Abs across domains of the SARS-CoV-2 Spike protein Omicron variant (A) BA2, (C) BA5. The numerical fractions above each bar indicate the number of ADCC-capable Abs, relative to the total number of Abs against the indicated domain. Of note, CTD Abs are clonally restricted, limiting our ability to draw conclusions.

Pairwise comparisons were performed using Fisher exact test with Holm correction for multiple testing. (B, D) ADCC activity of Abs by Spike domain Omicron BA2 (B), BA5 (D). Each point represents a single mAb. Boxplots show median, IQR, and 1.5 × IQR whiskers. Dotted line indicates limit of detection. Wilcoxon tests with Holm correction; significance indicated as *p < 0.05, **p < 0.01, ***p < 0.001, ****p < 0.0001; ns = not significant. All Abs were tested twice independently, except for four against BA.2 and one against BA.5, where only single measurements were available due to technical limitations (S1 Table).
(TIF)

**S2 Fig. ADCC activity elicited by S2-specific Abs stratified by epitope groups.** Each point represents one mAb, boxes show median and interquartile range, dotted line shows the limit of detection, fusion peptide is abbreviated FP. Epitope groups represented by a single antibody (n = 1) are displayed but were excluded from statistical testing. Overall differences among epitope groups were assessed by Kruskal-Wallis (A) D614G (p = 0.7109), (B), BA.2 (p = 0.6721) and (C) BA.5 (p = 0.2041); Dunn multiple comparisons were not significant.
(TIF)

**S3 Fig. Relationship between antigen binding and ADCC activity.** (A) Flow cytometry gating strategy for identifying mAb binding to SARS-CoV-2 Spike. Cells were sequentially gated on singlets, live cells, Spike+ population, and mAb binding was detected using a secondary anti-IgG Fc antibody. Blue dots represent the negative control, and red dots indicate the positive control mAb CV3-25 GASDALIE. (B, C) Correlation between ADCC activity and Spike binding (MFI) against BA.2 (B) and BA.5 (C) variants. Analysis includes Abs classified as binders (MFI > 1000). Each point represents one mAb. Pearson correlation coefficient (R), p-value, and 95% confidence interval (gray shading) from linear regression are shown.
(TIF)

**S4 Fig. Correlation between Abs maturation and neutralization activity.** Scatter plots show the correlation between Abs maturation %SHM, (A) VH; (B) VL) and neutralization activity ($\log_{10}$[$IC_{50}$, ng/mL]) against WH-1 pseudovirus among neutralizing Abs. Each point represents one mAb. Pearson correlation coefficients (R), p-values, and 95% confidence intervals (gray shading) from linear regression are shown.
(TIF)

**S5 Fig. Longitudinal comparison of Abs maturation and ADCC activity against Omicron Spike variant.** (A-B) %SHM in the VL (A) and VH (B)_of Abs at PVI-1 and PVI-2. (C, D) ADCC activity against Omicron BA.2 (C) and BA.5 (D) Spike variants at each time point. P-values from two-sided Wilcoxon rank-sum tests are shown. Boxplots indicate the median, interquartile range (IQR), and whiskers represent 1.5 × IQR. Dotted line indicates limit of detection.
(TIF)

**S6 Fig. Correlation between Abs CDR3 length and neutralization activity** Scatter plots show the correlation between CDR3 length VL (A), VH (B) and neutralization activity ($\log_{10}$[$IC_{50}$, ng/mL]) against WH-1 SARS-CoV-2. Analysis includes neutralizing Abs, as described in methods. Pearson correlation coefficient (R), p-value, and 95% confidence interval (gray shading) are shown.
(TIF)

**S7 Fig. ADCC assay gating strategy, controls, and reproducibility.** (A) Representative gating strategy used for the ADCC assay. (B) Representative ADCC activity of negative (A32, S2X259) and positive (S309, CV3-25 WT, CV3-25 GASDALIE) controls against WH-1 SARS-CoV-2 Spike. (C,) Correlation of ADCC activity between independent experimental replicates for WH-1 (left), BA.2 (middle), and BA.5 (right) Spike variants. Pearson correlation coefficients (R) and p-values are shown.
(TIF)

**S1 Table. Figures source data and raw data.**
(XLSX)

**S2 Table. Antibodies amino acid sequences.**
(XLSX)

## Acknowledgments

We would like to thank Helen Chu and the participants and the study staff of the Hospitalized or Ambulatory Adults with Respiratory Viral Infections study for providing the samples that were used as the source of monoclonal antibodies. We thank Andres Finzi for providing Spike-expressing cells and sharing methods. We thank Duncan Ralph for running Partis analysis.

## Author contributions

**Conceptualization:** Delphine M Depierreux.

**Formal analysis:** Delphine M Depierreux, Felicitas Ruiz, Michelle Lilly, Jamie Guenthoer, Vrasha Chohan.

**Funding acquisition:** Julie Overbaugh.

**Investigation:** Delphine M Depierreux, Felicitas Ruiz, Michelle Lilly, Jamie Guenthoer, Vrasha Chohan.

**Methodology:** Delphine M Depierreux.

**Supervision:** Julie Overbaugh.

**Visualization:** Delphine M Depierreux.

**Writing – original draft:** Delphine M Depierreux.

**Writing – review & editing:** Delphine M Depierreux, Felicitas Ruiz, Michelle Lilly, Jamie Guenthoer, Vrasha Chohan, Julie Overbaugh.

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
