## [Decision Letter · Decision Letter 0]

4 Jan 2026

PPATHOGENS-D-25-02145

Determinants of Natural Killer Cell-Mediated Antibody Dependent Cellular Cytotoxicity in SARS-CoV-2 Antibodies

PLOS Pathogens

Dear Dr. Overbaugh,

Thank you for submitting your manuscript to PLOS Pathogens. After careful consideration, we feel that it has merit but does not fully meet PLOS Pathogens's publication criteria as it currently stands. Therefore, we invite you to submit a revised version of the manuscript that addresses the points raised during the review process.

We look forward to receiving your revised manuscript.

Kind regards,

Christopher F. Basler

Academic Editor

PLOS Pathogens

Ashley St. John

Section Editor

PLOS Pathogens

Sumita Bhaduri-McIntosh

Editor-in-Chief

PLOS Pathogens

orcid.org/0000-0003-2946-9497

Michael Malim

Editor-in-Chief

PLOS Pathogens

orcid.org/0000-0002-7699-2064

**Additional Editor Comments:**

Your manuscript was viewed positively, but Reviewers 1 and 2 requested a number of clarifications to the text/figures and Reviewer 3 suggested that additional experiments might be warranted.

**Journal Requirements:**

1) We have noticed that you have uploaded Supporting Information files, but you have not included a list of legends. Please add a full list of legends for your Supporting Information files after the references list.

**Reviewers' Comments:**

Reviewer's Responses to Questions

**Part I - Summary**

Reviewer #1: In this manuscript, Depierreux et al. present a systematic and well-executed analysis of the determinants of antibody-dependent cellular cytotoxicity (ADCC) mediated by SARS-CoV-2-specific mAbs. Leveraging an unusually large and internally consistent panel of 142 mAbs isolated longitudinally from a single donor with hybrid immunity, the authors interrogate the relationship between ADCC activity and epitope specificity, antigen binding, neutralization, SHM, and CDR3 length. Their central conclusion that ADCC activity is primarily dictated by epitope targeting, particularly toward the S2 domain of the Spike protein, rather than by affinity maturation, binding strength, or neutralization potency is clearly supported by the presented data. The study is technically strong and the manuscript clearly written, and thoughtfully contextualized within the current literature. Its major strengths include the scale of the mAbs panel, the use of a uniform IgG1 Fc backbone to isolate Fab-driven effects, and the integration of functional, genetic, and binding data across multiple viral variants. Overall, this work represents a valuable contribution to our understanding of Fab-dependent control of Fc-mediated effector functions and has important implications for vaccine design and antibody-based therapeutics. However, there are a few minor conceptual and experimental limitations. These primarily relate to the generalizability of conclusions derived from a single donor, the reliance on domain-level rather than fine epitope mapping, and the interpretation of “lack of correlation” results given assay design constraints.

Reviewer #2: The authors present a study in which they investigate the mechanisms that govern antibody-dependent cellular cytotoxicity (ADCC) activity in neutralizing and non-neutralizing antibodies obtained from a vaccinated patient after breakthrough infections with delta and omicron variants. The authors find that ADCC activity was predominantly influenced by the region of the spike protein targeted by the mAbs, with the S2 domain giving rise to mAbs with the highest activity. Furthermore, ADCC activity did not correlate with neutralizing activity, binding affinity, or somatic hypermutation. These findings are novel and of importance to the academic community. The work is robust and is presented in a reasonable format; however, some modifications are necessary before publication.

Reviewer #3: This study analyzes 142 monoclonal antibodies isolated from a single patient at two time points for the ability to induce NK cell-mediated ADCC against SARS-CoV-2 S coated target cells. Monoclonal antibodies provide the advantage of being able to compare features of the IgG independent of titer and thus allow for comparison of Fab- and Fc-specific features. The authors have cloned all monoclonals on an IgG1 backbone, although I think it would be important to assess at least a subset for Fc-glycan composition to determine variation based on mAb production. The authors perform a number of comparisons across the mAbs, first comparing the epitope to which each mAb binds, and find that mAbs targeting the S2 domain and CTD have elevated ADCC activity compared to the other epitopes (RBD, NTD). For comparison, they include the neutralization data for the mAbs targeting each of those regions, where a majority of RBD and CTD binders are neutralizers whereas mAbs targeting NTD and S2 are predominantly non-neutralizing. Interestingly, the authors also analyze mAbs generated from B cells after a breakthrough infection from the same individual, and show that while neutralization potency increases, there was limited enhancement of ADCC-activity despite increased binding affinity and increased SHM.

The major strengths of the study are the focus on features of the Fab domain that impact ADCC potency through the use of monoclonal antibodies. While the authors report similar findings to prior reports (S2 domain-targeting antibodies can induce ADCC), the novelty is that they have controlled as best as they can for variation in the Fc domain by using mAbs all produced on an IgG1 backbone and assayed all mAbs at the same concentration, which ultimately allows for dissection of the contribution of different Fab features without the complicating factors of differences in titer or subclass distribution. Overall, I think that this is a nice study and fit for PLOS Pathogens, and it follows on the authors prior studies with this panel of mAbs.

**Part II – Major Issues: Key Experiments Required for Acceptance**

Please use this section to detail the key new experiments or modifications of existing experiments that should be absolutely required to validate study conclusions.required to validate study conclusions.

Reviewer #1: (No Response)

Reviewer #2: The authors use a flow cytometry-based method to determine which antibodies bind to spike-expressing cells. This is presented as mean fluorescence intensity. Based on this data, the authors conclude on lines 143-144 that “a minimum level of antigen binding is necessary for ADCC”. Does this assay actively measure affinity or just binding activity? A more robust measure of affinity would be provided by biolayer interferometry, which would provide association rate, dissociation rate, and equilibrium dissociation constant values.

The authors sequenced B cells from a single patient who received two mRNA vaccines and subsequently experienced two breakthrough infections with the delta and then the omicron variants. Did the patient receive any more vaccinations between these events? It would be useful for the reader to know the isotype of the identified mAbs. Each of the mAbs has been cloned into IgG1 backbones, ensuring comparable ADCC activity; however, it has been demonstrated that multiple vaccinations with SARS-CoV-2 mRNA vaccines elicit decreased IgG3 and increased IgG4 responses. Each of these mAb subtypes engages with CD16 to varying extents; therefore, the correlations between binding strength, neutralizing activity, and ADCC activity may differ in vivo from those observed in in vitro experiments. This data should be provided and elaborated on in the discussion section.

Reviewer #3: As Fc-glycan composition can impact binding to FcgR3A, it is important to assess the Fc glycan composition of at least a subset of mAbs to determine if there are batch to batch production differences in glycans. I am mostly interested in determining the fucose and sialic acid content, which could be assessed by lectin blots, across a subset of RBD, S2, and CTD binding mAbs.

**Part III – Minor Issues: Editorial and Data Presentation Modifications**

Reviewer #1: 1. A major strength of the study is the deep characterization of a large antibody repertoire from a single individual is also its most significant limitation. All antibodies are derived from one donor (C68) with a specific immunological history (vaccination plus two breakthrough infections). While this design minimizes inter-donor confounding and allows elegant internal comparisons, it remains unclear to what extent the observed dominance of S2-targeting antibodies in ADCC is a generalizable feature of human SARS-CoV-2 immunity versus a donor-specific immunodominance pattern. The authors briefly acknowledge this limitation, but the conclusions particularly those framed around general principles of ADCC determinants would benefit from a more explicit discussion of how repertoire bias, HLA type, exposure sequence, or germline gene usage might influence these findings. Even limited comparison to previously published monoclonal datasets from additional donors (where available) would help contextualize the broader applicability of the conclusions.

2. The manuscript convincingly demonstrates that mAbs targeting the S2 domain are enriched for ADCC activity relative to RBD- or NTD-directed antibodies. However, epitope assignment is performed at the level of broad Spike domains rather than finer structural or functional epitopes. Given the substantial heterogeneity within S2, including fusion peptide-proximal regions, heptad repeats, and stem helix epitopes, it is possible that only specific sub-regions within S2 are optimal for ADCC induction. This limitation is particularly relevant to the mechanistic interpretation proposed in the Discussion, where membrane proximity and synapse geometry are invoked as potential explanations. Without higher-resolution epitope mapping (e.g., competition groups, alanine scanning, or structural data), it remains speculative whether all S2 epitopes are equally competent for ADCC. The authors should clarify this point and temper conclusions accordingly, or explicitly state that future work will be required to resolve sub-domain effects.

3. The conclusion that antigen binding strength does not correlate with ADCC activity is intriguing and consistent with prior observations in HIV and influenza systems. However, binding in this study was assessed by single-concentration flow cytometry (MFI at 5 µg/ml), which primarily reports surface engagement rather than affinity, avidity, or kinetics. As such, the statement that “binding does not correlate with ADCC” should be interpreted more narrowly as “binding at a fixed concentration does not predict ADCC magnitude.” This distinction is important, as kinetic parameters (on-rate, off-rate) and antigen mobility on the target cell surface are increasingly recognized as contributors to Fc receptor clustering and signaling. The authors should clarify the limitations of their binding assay and avoid over-interpreting these data as evidence that affinity or avidity are irrelevant to ADCC.

4. ADCC is defined here as NK cell degranulation (expression of CD107a), which is an accepted and widely used surrogate readout. However, CD107a mobilization does not necessarily equate to target cell killing, nor does it capture the full spectrum of Fc effector functions. While the consistency of results across variants supports robustness, it would be useful to explicitly acknowledge that the conclusions relate to NK-cell activation rather than direct cytotoxic outcomes. In particular, readers may be tempted to extrapolate these findings to in vivo protection or therapeutic efficacy. Clarifying the scope of inference, i.e. NK activation versus killing or viral control, would improve data interpretation.

5. The CTD-specific antibody group is very small and clonally restricted, as acknowledged by the authors. It would be helpful to more clearly de-emphasize CTD-related conclusions throughout the text, including figure legends, to avoid over-interpretation.

6. The statistical analysis is adequate and presentation is generally clear, but several correlation analyses rely on Pearson’s R despite non-normal distributions. While this does not substantially affect conclusions, brief justification or confirmation that Spearman correlations yield similar outcomes would strengthen rigor.

Reviewer #2: In figures 1D, 1F, 2A, 2C, 5B, S1B, S1D, S4C, and S4D, limits of detection should be added to the graphs. Presently, it is difficult to determine which mAbs have no ADCCC/neutralizing activity.

Reviewer #3: 1. Please include details in materials and methods about which cell line/expression system was used to produce the mAb panel.

2. What was the circulating strain at the time of each breakthrough infection? This seems important to clarify and understand the findings in Fig S4C, where mAbs isolated after breakthrough 2 have reduced ADCC against BA.2 compared with breakthrough 1, but no difference in ADCC against BA.5. The authors should provide some additional discussion and/or hypotheses to explain these results.

PLOS authors have the option to publish the peer review history of their article (what does this mean? ). If published, this will include your full peer review and any attached files.). If published, this will include your full peer review and any attached files.

**Do you want your identity to be public for this peer review?** For information about this choice, including consent withdrawal, please see our For information about this choice, including consent withdrawal, please see our Privacy Policy ..

Reviewer #1: No

Reviewer #2: No

Reviewer #3: **Yes:** Bronwyn GunnBronwyn Gunn

**Figure resubmission:**
---

## [Editor Report · Decision Letter 1]

23 Feb 2026

Dear Dr. Overbaugh,

We are pleased to inform you that your manuscript 'Determinants of Natural Killer Cell-Mediated Antibody Dependent Cellular Cytotoxicity in SARS-CoV-2 Antibodies' has been provisionally accepted for publication in PLOS Pathogens.

Best regards,

Christopher F. Basler

Academic Editor

PLOS Pathogens

Ashley St. John

Section Editor

PLOS Pathogens

Sumita Bhaduri-McIntosh

Editor-in-Chief

PLOS Pathogens

orcid.org/0000-0003-2946-9497

Michael Malim

Editor-in-Chief

PLOS Pathogens

orcid.org/0000-0002-7699-2064
---

## [Editor Report · Acceptance letter]

Dear Dr. Overbaugh,

We are delighted to inform you that your manuscript, "Determinants of Natural Killer Cell-Mediated Antibody Dependent Cellular Cytotoxicity in SARS-CoV-2 Antibodies," has been formally accepted for publication in PLOS Pathogens.

Best regards,

Sumita Bhaduri-McIntosh

Editor-in-Chief

PLOS Pathogens

orcid.org/0000-0003-2946-9497

Michael Malim

Editor-in-Chief

PLOS Pathogens

orcid.org/0000-0002-7699-2064